# The Burden of Next-of-Kin Carers of Dementia Sufferers in the Home Environment

**DOI:** 10.3390/healthcare10122441

**Published:** 2022-12-03

**Authors:** Miroslava Tokovska, Jana Šolcová

**Affiliations:** 1Department of Health and Exercise, School of Health Sciences, Kristiania University College, Prinsens gate 7-9, 0152 Oslo, Norway; 2Department of Social Work, Faculty of Education, Matej Bel University, Ruzova 11, 974 11 Banska Bystrica, Slovakia

**Keywords:** next-of-kin, carers’ stress, relative stress scale, dementia sufferers, Slovakia

## Abstract

The role and responsibilities of next-of-kin carers are significant, filling several years of their lives and causing them to experience a burden of care. This study was conducted to investigate the burden of next-of-kin carers for dementia sufferers in Slovakia. Data were compiled via the Relatives’ Stress Scale (RSS) screening instrument through a survey of 112 primary next-of-kin carers and analysed using the statistical tests of descriptive statistics, means, scores and significance differences (Wilcoxon test). This is one of the few scales which provides: (a) specific measures of caregiver stress, and (b) is standardised for a population of informal carers in the home environment. The profiles of Slovak next-of-kin carers were identified with statistically significant characteristics (gender, age of carers). The study identified a high level of caregiving stress (82.15%), and selected factors were found to be significant in all burden dimensions: emotional stress (*p* = 0.001), social stress (*p* = 0.003), and negative feelings (*p* = 0.002). The results emphasise the need for coordination of healthcare and social services, possibly by expanding the network of social support groups, operating a counselling hotline/chat, and introducing national educational programmes for the next-of-kin carers of dementia sufferers. The results are also a source of reference for the umbrella organisation ‘the Slovak Alzheimer’s Society’ to access when implementing changes at a national level.

## 1. Introduction

The position of informal carers is one of the key issues in dementia care. Informal care forms a cornerstone of all long-term care systems in Europe and has been gaining increasing recognition in international policy circles as a key issue for future welfare policy. The European Pillar of Social Rights (2017) explicitly makes a commitment to people providing care, highlighting their right to flexible working and access to care services. Since the caring function of families remains the key type of care provision, informal care is often seen as a cost-effective way of preventing institutionalisation and enabling users to remain at home [1]. The needs of carers and the impact of providing informal care on key life outcomes such as employment, health, and well-being are further being increasingly recognised in academic literature and in national policies across Europe [2,3]. The number of informal carers for family members suffering from dementia is growing worldwide regardless of ethnicity, social status, age, or gender [4,5,6]. Next-of-kin carers can be defined by the relationship they have with the sufferer (spouse, adult children, daughters-in-law and sons-in-law, friends, neighbours), living arrangements (co-resident with the care recipient or living separately) and care input (regular, occasional, or routine). Next-of-kin as informal carers are involved in providing unpaid ‘hands-on’ care or in organising care delivered by others, sometimes organising it from a distance [7,8].

The major burden of care and support for dementia sufferers usually falls on one person, commonly the next-of-kin, who takes on the role of the main domestic caregiver. As a result, this caregiver often faces considerable hardship in terms of the physical, emotional and social burden, along with experiencing financial stressors and negative feelings. Several studies report that a considerable amount of informal care is provided in the home environment, ranging from 30 to 100 hours per week. Women have been identified as the main informal caregiver for 55–91% of dementia sufferers [6,7,8,9]. Caring for the next-of-kin of dementia sufferers can be experienced as stressful and associated with negative effects on health, both the psychological and somatic morbidity of informal carers [10,11,12,13]. 

The population data of Slovakia show that there is a slight increase in the population forecast for the period 2018 and 2025, with a decrease between 2025 and 2050. Despite these marginal changes, the overall number of dementia sufferers will more than double from 62,495 in 2018 to 128,986 in 2050 [14] (p. 79). Similarly, as a percentage of the overall population, dementia sufferers will represent 2.59% in 2050 compared to 1.15% in 2018, meaning Slovakia exceeds the broader European trend for the number of dementia sufferers. With the growing number of dementia sufferers, the need for both formal and informal care will increase. Several Slovak studies describe various determinants of stress for those caring for dementia sufferers, and also family caregiving involving end-of-life care [15,16]. Slovak research into the consequences of care has disclosed that 53% of next-of-kin reported a psychological burden resulting from the custody of relatives, and 44% of next-of-kin additionally report a burden relating to physical care [15,17,18]. Slamkova et al. (2020) [19] pointed to the increase in the burden on next-of-kin carers in care activities and changes in their state of health in relation to their emotions. According to Poliakova et al. (2020) [20], who conducted research based on self-designed questionnaires for informal long-term care for the elderly with dementia in Slovakia, informal carers lack the theoretical knowledge and practical skills that are necessary for the implementation of quality and specific care for the elderly with dementia. 

There is a need to understand the profile of the next-of-kin carers and their role as informal carers of dementia sufferers in order to gain useful knowledge that can be operationalised when planning future dementia care. In collaboration with the Slovak Alzheimer’s Society, a screening survey was conducted with a focus on three burden dimensions, using the standardised Relatives’ Stress Scale: (a) emotional distress; (b) social distress; and (c) negative feelings. Thus, this study aims to investigate the burden of next-of-kin carers of dementia sufferers in Slovakia by screening and analysis of these dimensions of stress through the research question: What is the burden of the next-of-kin carers of dementia sufferers in Slovakia according to the RSS screening instrument?

## 2. Materials and Methods

### 2.1. Study Design

This study has a quantitative and descriptive design. The next-of-kin carers’ burden was measured with the Relatives’ Stress Scale (RSS) [21]; according to Lesher (2015) [22], due to the numerous definitions and operationalisations of caregiver stress, there are many different instruments designed to measure this construct, and the RSS is one which is used to measure multiple dimensions of subjective caregiver stress. The Relatives’ Stress Scale is one of the few identified that is able to: (a) specifically measures caregiver stress, and (b) is normed on a population of informal carers for individuals with dementia [21]. This scale is widely used in both clinical practice and research [23] and is especially useful because its subscales allow the examination of different dimensions of carers’ stress, such as ‘emotional distress’, ‘social distress’ and ‘negative feelings’ [24]. The RSS has demonstrated good construct validity [25], and reliability [21]. This RSS questionnaire can be used for research on stress among informal carers with public-accessed permission from the Norwegian National Advisory Service on Ageing and Health, the latest revision being RSS 2019 (intern no. 00007) [26]. In Norway, RSS was used repeatedly to screen the burden of next-of-kin [27,28]. It should be noted that there are other screening assessments that could be used, such as those described in the publication ‘Screening for Stress and Burden in Carers of Seniors’ [29]; a burden scale such as the RSS, however, focussing on the general problems associated with the caregiving role, is a suitable screening instrument for our study. The questionnaire is quick to complete and can be used by spouses as well as other family carers. Ulstein et al. (2007a) [30] also stressed that RSS can be used as a useful basis for discussions with family carers. 

Since RSS is not carried out ordinarily in Slovakia, it was necessary to translate and adapt the scale for the Slovak language and culture by the first author (MT), and then for this adaptation to be checked and revised by the second author (JS). As part of the process of screening the stress of next-of-kin carers, we captured the problems of perform ing the role of caregiver in the home environment using our translated standardised Relatives’ Stress Scale questionnaire, using three dimensions: (a) emotional distress (items 1–6); (b) social distress (items 7–13 without 12); and (c) negative feeling (items 12, 14–15) [22,24,25,30]. As part of the statistical investigation, we found statistically significant areas in all dimensions. Another part of screening involved the coherence between the selected characteristic (factors) of next-of-kin carers and dementia sufferers, and the subjective perception of a specific need. The scale consists of 15 items, each rated from 1 to 4 (1 = not at all, 2 = rarely, 3 = frequently/quite a lot, 4 = always). A score above 23 on the RSS indicates an increased risk of clinically significant sociological and psychological distress [24]. 

One observation we made during the theoretical analysis of RSS as the screen instrument is the discovery of different approaches to defining the term ‘relative’. While Scandinavia uses the term ‘relatives’ stress scale [24,26,27,28,30], which can only be taken as meaning ‘a family member’, Anglo-Saxon and eastern cultures use the term ‘relative’ stress scale [31,32,33], presumably meaning that the stresses are in comparison—relative—to each other, thus focussing the attention on the role of the relative and the stress they feel [34]. In this study, we have used the term ‘RSS’ to mean ‘Relatives’ Stress Scale’.

### 2.2. Sample

An online and printed questionnaire was sent to 150 participants through five support groups in Slovakia, coordinated by the Slovak Alzheimer’s Society. Support groups for relatives of patients with Alzheimer’s disease and dementia aetiology of various types provide their members with a safe space for mutual discussions and the sharing of experiences, as well as for social and emotional support. In Slovakia, there are five support groups focussing on the close relatives of persons with Alzheimer’s disease [28]. Since support groups have open, voluntary and anonymous membership, there is no directory of totally registered support group members. 

### 2.3. Data Collection

In November 2019, a request to assist in the distribution of questionnaires was sent to the leaders of support groups registered by the Slovak Alzheimer’s Society. A total of 112 next-of-kin dementia sufferers in Slovakia responded to the request from December 2019 to February 2020.

The leaders were instructed to inform potential participants about the study and give them an envelope containing written information about the study, a consent form, and the questionnaire. The questionnaire could optionally be completed via a paper version (60.2% of the analytic sample) or by a web-based version (39.8%). Participants were instructed that the answers they gave in the questionnaire should exclusively concern the care they provide to their dementia sufferer. There were two inclusive criteria that needed to be fulfilled: (1) to be next-of-kin, and (2) to be caring for dementia sufferers. 

Enclosed with the online and printed questionnaire was a cover letter informing the next-of-kin carers about this research study and ensuring the anonymity and confidentiality of both the carers and their data. Contact details were given in the contact letter in case carers required further information before deciding to participate, or if they wanted to be kept informed of the results of the study. Next-of-kin carers were free to return the questionnaire to the leader of the support group, or not if they preferred, and they could withdraw their data from the study at any time. Return of the questionnaire, however, implied consent to participate in the study.

### 2.4. Ethical Approval

Before the research started approval was gained from the relevant Ethics Commission by Matej Bel University, Department of Social Work (MBU-FoE-DSW 14/2019). All participants received an explanation of the purpose of the study and agreed to participate by signing informed consent via a printed survey. Participants in the online survey agreed that the return of the questionnaire implied consent to participate in the study. The study strictly adheres to the ethical principles of the Declaration of Helsinki [35] and the provisions of the Oviedo Convention [36].

### 2.5. Data Analysis

The questionnaires yielded quantitative data. The reliability of the data was determined by recalculating the averages, median and score, and by monitoring the statistical significance of the differences (the result of the Wilcoxon test (two related samples)—level of significance = *p*). From the study, the results that came out as statistically significant are presented. For the level of significance (*p*-sig.) of the used test, a value of 0.05 was determined (if the SPSS output was 0.000, the form *p* < 0.001 was written) [37]. 

## 3. Results

### 3.1. Profile of Next-of-Kin Carers

The background characteristics of next-of-kin carers and dementia sufferers are given n Table 1. In total, 112 of 150 (75%) next-of-kin carers we approached took part in this study. The results in the table show the age of the research participants. Descriptive indicators in the sample of participants pointed to several specific situations with relative carers. Table 1 shows that significantly more women take care of dementia sufferers (*p* < 0.001). Dementia sufferers’ range in age from 52 to 97 years, while next-of-kin carers range in age from 43 to 88 years. There is a small difference in the average age of dementia sufferers (M = 78.88) and next-of-kin carers (M = 74.65). Almost 30% of the next-of-kin carers are children of the sufferers, and the majority of them are women (65.7%). With regard to the number of hours spent caring per week, no significant differences were found between those carers who were externally employed and those who were unemployed. Spouses and partners are most represented in the position of closest relative carers (together, 38.4%). Many of these carers would themselves need support and help/care due to their age (30.45% of them are aged 70+). Under Other, specific relatives and non-relatives included: godson, uncle, brother, sister, and one family acquaintance.

According to Ulstein et al. (2007) [30], a score above 23 on the RSS indicates an increased risk of clinically significant socio-psychological distress. Our results clearly show a high level of caregiving stress—82.15% within the range of 24 to 60—whereas a low level of stress—with a score of 23 or lower—is 17.85%. Of further interest is the fact that 10.8% of the total responses indicating caregiving stress scored the maximum available on the scale (60). The youngest was a 27-year-old granddaughter providing care to her grandfather (score = 60) and the oldest was a daughter at the age of 61 providing care to her father (score = 60). In the context of the lower scores, we consider it interesting that the lowest score was 8—in this case, the next-of-kin at the age of 75 stated that she provides care to her brother at the age of 77 and that they are both retired.

The mentioned findings helped us to identify selected factors (characteristics) of next-of-kin carers, in which we focussed on searching for connections with individual dimensions of burden.

### 3.2. Selected Factors Related to the Burden Dimensions of Next-of-Kin Carers Caring for Dementia Sufferers

Burden dimensions of the RSS resulted in three subgroups: (i) Emotional distress (six items); (ii) Social distress (six items); (iii) Negative feelings dimension (three items) (more in 2.4). The level of significance of selected factors influencing the subjective perception of the burden on next-of-kin carers in the subscale ‘Emotional distress’ is described in Table 2.

In the ‘Emotional distress’ dimension (Table 2), statistically significant differences were noted in all monitored factors regarding: (a) next-of-kin carers—gender, age, relationship to sufferer; and (b) the dementia sufferers—their age.

Within the factor ‘gender’ of next-of-kin carers, we found statistically significant differences in subjective perception in responses to these questions: ‘Do you ever feel you can no longer cope …?’ (*p* = 0.003), ‘Do you worry about potential accidents …?’ (*p* = 0.008), and ‘Do you ever get depressed by the situation?’ (*p* = 0.002). Responses to ‘Do you worry about potential accidents …?’ produced a Kurtosis (skewness) figure of 2.0 ≥ 0, meaning that most of the values in the observed set were close to the mean. It was a rectangular division, i.e., the closest relatives ticked several higher values to the right of the mean. Within the factor ‘Age of next-of-kin caregiver’ we found statistically significant differences in subjective perception in responses to ‘Do you ever feel you can no longer cope …?’ (*p* = 0.001) and ‘Do you ever get depressed by …?’ (*p* = 0.002). In the factor ‘Relationship’, we identified significant associations in the responses to the questions ‘Do you ever feel that you can no longer cope …?’ (*p* = 0.040), ‘Do you ever feel you need a break?’ (*p* = 0.030), and ‘Do you ever feel that there will be no end…?’ (*p* = 0.003).

Within the factor ‘Age of dementia sufferers’, we identified a significant need for respite care in the responses to the question ‘Do you ever feel you need a break? ‘(*p* = 0.001).

Within the dimension of social need (Table 3), we noted statistically significant differences in the subjective assessment of the level of development of specific abilities and skills in three out of four selected factors. However, for all of them the average rating was higher than the mean: M ≥ 3.50 (range 3.57 to 4.19). A statistically significant difference in the factor ‘Gender of the next-of-kin carers’ was identified. An association with the question ‘How much has the domestic routine…?’ was demonstrated here (*p* = 0.013). Within the age of next-of-kin carers, statistically significant differences were found in the subjective perception of the responses to ‘Is your sleep interrupted by…?’ (*p* = 0.001).

The age factor of dementia sufferers statistically significantly determined the carers’ response to the questions ‘Do you find it difficult to get …’ (*p* = 0.001), and ‘How much has the domestic routine been …?’ (*p* = 0.002).

Table 4 shows that all four factors were identified during the statistical investigation. In this case, the factor ‘Age of the closest carers’ is filled with several items. Statistical significance was demonstrated by the statement ‘Do you ever feel embarrassed by…?’ (*p* = 0.001) and the statement ‘Do you ever get cross or angry with…?’ (*p* = 0.002). A statistically significant difference was identified in the gender of the closest relatives of carers in the statement ‘Do you ever feel frustrated with…?’ (*p* = 0.002). The factor ‘Age of dementia sufferers’ was statistically significantly determined by the closest relative of the caregiver in the statement ‘Do you ever feel frustrated with…?’ (*p* = 0.007). In relationships, there appeared significant connections with the statement ‘Do you sometimes feel embarrassed by…?’ (*p* = 0.003).

Based on the results, it is possible to state the existence of a statistically significant difference in terms of the age of dementia sufferers (*p* = 0.003).

## 4. Discussion

The primary goal of this study was to screen and analyse the profile and burden dimensions of the next-of-kin carers for dementia sufferers in Slovakia using the Relatives’ Stress Scale (RSS) screening instrument. The self-administrated RSS covers various aspects of burden, such as subjective emotional responses, restrictions on the social life of the next-of-kin, and negative feelings associated with the dementia sufferers and their behaviour. The result of the factor analysis turned out to be identical to those performed 40 years ago in Britain [20], 15 years ago in Norway [23], and recently in Germany [38] and Thailand [32], which displays the usefulness of the scale over time and across cultures. Being a next-of-kin carer of someone suffering from dementia of any aetiology is stressful [15,16,17,18,19,39,40,41,42,43,44]. The results of this study are well-aligned with the existing research that suggests some carers experience a great deal of burden; this is also the case for Slovak next-of-kin carers for dementia sufferers. Brodaty and Donkin (2022) [43] described that family carers for dementia sufferers, often called the invisible second patients, are critical to the quality of life of the care recipients. Several studies and review articles have demonstrated higher rates of psychiatric symptoms (such as depression and anxiety) among carers, especially among women, and confirmed that the carers are so overwhelmed by their caregiving responsibilities that they are likely to have (or soon develop) symptoms severe enough to meet the full criteria for a psychiatric disorder [45,46,47,48]. Prior research with dementia carers [28] found that stress, as measured by the RSS, was a powerful discriminatory factor regarding the risk for caregiver psychiatric illnesses such as depression. 

A World Health Organisation report (2021) [46] highlights the urgent need to strengthen support at the national level, in both formal and informal settings—both in terms of care for dementia sufferers and in support for the people who provide that care. Furthermore, this report stressed that primary health care should be included for dementia sufferers, along with specialist care, community-based services, rehabilitation, long-term care, and palliative care. Alzheimer Europe also calls on its member countries to formalise and implement national dementia plans with priority outlines in the long-term planning of healthcare and social services for dementia sufferers and their informal carers. Many countries aligned to Alzheimer Europe, such as Norway, have already done so; the Norwegian government and local authorities have used available data from research studies on the stress levels of next-of-kin to expand and improve the portfolio of existing services. This also includes new services for their citizens suffering from dementia, with a focus on next-of-kin. The Slovak Alzheimer’s Society is also a member of the Alzheimer Europe organisation, but Slovakia is one of the few European countries which does not have a national dementia strategy [47]. Without this important national strategy, and without the specifications of outlined focus areas, it is difficult to implement dementia policy to help carers and dementia sufferers. The lack of a creative solution to this demand among Slovak next-of-kin carers may lead to permanent damage to health for next-of-kin carers and dementia sufferers. Ultimately, public health costs will increase. In our study, the average age difference between those providing care and those receiving the care was shown to be minimal. This implies that older adults are taking care of the elderly, although they themselves would increasingly need help; in turn, the need for geriatric care rises for both groups. The consequences could be that Slovak primary and specialist healthcare and community-based services will have big challenges during the next ten years. 

Informal carers are often women, providing care to a spouse, parents or parents-in-law, and a large share is provided by people who are older than the standard retirement age [48,49]. Almost thirty per cent of the relatives of the next-of-kin carers who completed our research questions were children of the sufferers, the majority of them women (65.7%). Women are more often responsible for taking care of their own families and children and are often also externally employed, leading to a double burden and increasing the likelihood of negative health effects, potentially ending up in psychiatric treatment [50]. Future demographic changes, healthcare advances, long-term care policy, and cost-containment pressures will all bring about the favouring of community and informal care options over institutionalisation wherever possible [51].

One of the important factors that can contribute to the exhaustion of family members is their moral responsibility for caring for close or distant families. Poor coordination of social and health services, or a lack of services in rural areas, also increases the stress of caregiving. Slovakia is a collectivist country with a long-lasting Christian culture and history; the fundamental moral value, such as the duty of an adult child to take care of a sick and ageing parent, is strongly rooted in Slovak society. This can put pressure on the next-of-kin to take responsibility for the family member with dementia, thus increasing the carer’s emotional and social distress, together with the accompanying negative feelings. Hansen and Tran (2019) state [52] that the duty of adult children to their parents is an important aspect of morality in collectivist societies. Institutionalising parents with dementia can cause feelings of guilt and shame, and fears of stigmatisation and ostracism. 

To avoid blame and denial, carers try to keep the fact that family members have dementia ‘in the family.’ Such an attitude may increase the stress of next-of-kin carers. Family care morality may constitute a significant barrier against seeking professional help for dementia sufferers, a barrier based on the expectation that the family will care for their elderly, even when the dementia of those being cared for is severe. According to the WHO [50] (pp. 26–27), carers are involved in providing ‘hands on’ care and support for dementia sufferers or play a significant role in organising the care delivered by others. Informal carers often know the dementia sufferers well, and therefore are likely to have knowledge of and information about the person with dementia that is crucial for developing effective personalised needs-based treatment and care plans. Carers should therefore be considered essential partners in the planning and provision of care in all settings according to the wishes and needs of the person with dementia, and they should have access to the support and services tailored to their needs to effectively respond to and manage the physical, mental, and social demands of their caring role. Listening to the voices of the next-of-kin carers as informal carers of dementia sufferers—and increasing, improving, and adapting community-based services—will contribute to reducing the burden [53,54].

This study conducted on Slovakia’s next-of-kin carers’ dimensions of burden has some strengths and limitations. To the best of our knowledge, this is the first study that provided evidence using the screening instrument ‘Relatives’ Stress Scale’ (RSS). This study presents a snapshot of the situation in Slovakia and could be transferable to other countries. The significance of this study lies in providing baseline data and implications for improving existing policies, and developing new policies, to establish tailored psychosocial support as a part of social services or the healthcare system for next-of-kin carers of dementia sufferers. The collaboration with the Slovak Alzheimer’s Society was advantageous for this study and in turn motivated us, the researchers, to carry out this survey. Furthermore, our study fulfils the participatory action research (PAR) which is based on reflection, data collection, and action that aims to improve health and reduce health inequities by involving the people who, in turn, take actions to improve their own health. For instance, a burden dimension assessment included engagement with local communities through support groups from Slovakia and included a survey of residents.

There are also important limitations to note. The sample was limited to participants of support groups for next-of-kin carers for dementia sufferers. Not all relatives attend support groups, and the groups usually only exist in urban areas. For those living in rural areas, such opportunities are rare or non-existent. Furthermore, although this study identified the increased burden experienced by the groups of carers in Slovakia, it has not been established whether factors such as the severity of the disease, the engagement of caring services for community-based long-term care, and economic costs of dementia sufferers could impact the burden of these carers; these factors would benefit from further investigation. In our study, there is no information on the participants’ psychiatric morbidity or treatment for depression or anxiety. There is an assumption that some family members have undergone or are undergoing such treatment, which may affect the validity of the collected data. In addition, as part of the RSS validation, other statistical tests such as those described in the Section 2 ‘Material and Methods’ were not used. Finally, this study was carried out before the COVID-19 pandemic; in the current climate, the same survey could yield different results. Future research should be conducted on the use of other types of standardised screening tools, and on experiences in connection with stress and psychosocial burden by using dyadic interviews or focus group interviews. 

## 5. Conclusions

The study showed that caregiver burnout in the closest relative is the result of the accumulated burden during the caregiving process. This study also revealed that psychosocial support for the closest carers of dementia sufferers is important. The results emphasise the need for coordination of services for Slovak carers. This could be made possible by expanding the network of social support groups, operating a counselling hotline or chat, introducing national education programmes for next-of-kin, creating positions for the role of local communities’ coordinators, and increasing the connectivity and efficiency of the work of individual support groups throughout Slovakia. Such targeted activities are overseen by non-profit organisations, such as the Slovak Alzheimer’s Society. 

Targeted public health policy means legal, social, and political support, such as the development of a national strategic plan to support families with dementia, which the current government is trying to develop in cooperation with experts and non-profit organisations. Raising awareness about the challenges of informal carers, funding services to enhance availability, and training for informal carers—along with facilitating social relationships through peer support and self-help—would be significant support measures that address the needs of next-of-kin for dementia sufferers. Advocating for the mental health support of close family carers who are directly involved in the care of dementia sufferers requires further research to address the current lack of information on the health screening behaviours of this population. Based on research evidence, it is possible to plan targeted support services for the entire spectrum of family carers, not just for families with dementia.

## Figures and Tables

**Table 1 healthcare-10-02441-t001:** Demographic characteristics of next-of-kin carers and dementia sufferers.

Variable		
Gender of next-of-kin carers	Woman	83.80%
	Men	16.20%
Age of dementia sufferers	Mean (SD), range	78.88 (7.70) 52 to 97
Age of next-of-kin carers	Mean (SD), range	74.65 (6.80) 43 to 88
Relationship	Children	29.90%
	Partners	15.10%
	Grandchildren	23.40%
	Family in law	23.30%
	Other	8.30%

**Table 2 healthcare-10-02441-t002:** Differences in the RSS evaluation based on selected factors.

		Gender of Next-of-Kin Carers	Age of Dementia Sufferers	Age of Next-of-Kin Carers	Relationship
Emotional Distress	M	*p*	*p*	*p*	*p*
1. Do you ever feel you can no longer cope with the situation?	3.79	0.003		<0.001	0.040
2. Do you ever feel you need a break?	3.88		<0.001		0.030
3. Do you ever get depressed by the situation?	3.96	0.008		0.002	
4. Has your own health suffered at all?	3.60				
5. Do you worry about potential accidents happening to…?	4.17	0.002			
6. Do you ever feel that there will be no end to the problem?	3.46				0.003

RSS = Relatives’ Stress Scale; M = mean, *p* = value.

**Table 3 healthcare-10-02441-t003:** Differences in the RSS assessment based on selected factors.

		Gender of Next-of-Kin Carers	Age of Dementia Sufferers	Age of Next-of-Kin Carers	Relationship
Social Distress	M	*p*	*p*	*p*	*p*
7. Do you find it difficult to get away on holiday?	4.13		<0.001		
8. How much has your social life been affected?	4.15				
9. How much has the domestic routine been upset?	4.19	0.013	0.002		
10. Is your sleep interrupted by …?	3.72			<0.001	
11. Has your standard of living been reduced?	3.77				
13. Are you at all prevented from having visitors?	3.57				

RSS = Relatives’ Stress Scale; M = mean, *p* = value.

**Table 4 healthcare-10-02441-t004:** Differences in RSS evaluation based on selected factors.

		Gender of Next-of-Kin Carers	Age of Dementia Sufferers	Age of Next-of-Kin Carers	Relationship
Life Upset	M	*p*	*p*	*p*	*p*
12. Do you ever feel embarrassed by…?	3.59			<0.001	0.003
14. Do you ever get cross or angry with…?	3.22			0.002	
15. Do you ever feel frustrated at times with…?	3.37	0.007	<0.001		

RSS = Relatives’ Stress Scale; M = mean, *p* = value.

## Data Availability

Not applicable.

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
