# Peer review of "The Burden of Next-of-Kin Carers of Dementia Sufferers in the Home Environment"

_healthcare, 2022, doi:10.3390/healthcare10122441_

Round 1

Reviewer 1 Report

This is an interesting study that may have potential clinical implication at national level.

There are some weaknesses of the study and some information is missing.

There is no information about the total number of persons that were initially approached for participation in the study. It would be interesting to know the rate of participation, and whether it is similar with other research, given that all participants attended a support group. Are there any data about possible differences between carers that attend a support group and those who do not? Their profile may differ. Moreover, given the advanced age of most caregivers, one could hypothesize that many more might have had difficulties in participating in this study.

There is no information regarding the potential support and other resources the caregivers had in this study, that could alleviate their burden. Also, no information regarding the hours that the caregivers spent in the care of dementia patients.

There is no information with regards to psychiatric morbidity and medication of the participants. The authors state that caregiving is associated with depression and anxiety, so it would be expected a number of participants to have been treated for those disorders. It seems plausible that a person who had previously clinically relevant symptoms, possibly related to the caregiving status, may have score better in this scale, only because he/she receives and antidepressant or anxiolytic. Those issues should be commented by the authors.

In the results section (lines 169-171) the authors refer to unemployed carers. However, given the mean age of carers (>74 years) it would be expected that most would have been retired. Please clarify.

The most part of the first paragraph in the Discussion section (lines 253-259) should be removed to Methods section.

The authors state that this is the first study in their country with the use of the RSS. Are there other studies in Slovakia on the same topic that used other instruments? If so, you can make some general comparisons with your results.

Author Response

This is an interesting study that may have potential clinical implication at national level.

Thank you for the positive feedback.

There are some weaknesses of the study and some information is missing.

There is no information about the total number of persons that were initially approached for participation in the study. It would be interesting to know the rate of participation, and whether it is similar with other research, given that all participants attended a support group. Are there any data about possible differences between carers that attend a support group and those who do not? Their profile may differ. Moreover, given the advanced age of most caregivers, one could hypothesize that many more might have had difficulties in participating in this study.

We added information about the sample - a questionnaire was sent to 150 informants. Unfortunately, we do not have data on the profile and possible differences between caregivers who attend a support group and those who do not. Since there are no accurate registers of patients and their caregivers in Slovakia, it is not possible to collect this data. Support groups do not register members, because the main principles of participation in a support group are: voluntary, anonymous and free entry. Another principle is to feel free and safe to talk to other members. Members don't have to reveal their names, just talk about their experiences or ask other members what they want to learn or share. We agree with the reviewer's assumption that, in relation to participation in our study, several carers may have had difficulty participating due to their advanced age. Given that this is a study with the aim of screening and analyzing the stress dimensions of the next-of-kin carers of dementia sufferers in Slovakia, which was carried out using the Relatives' Stress Scale (RSS) screening tool, we contacted informants who participated in the support group during a specific period.

There is no information regarding the potential support and other resources the caregivers had in this study, that could alleviate their burden. Also, no information regarding the hours that the caregivers spent in the care of dementia patients.

We understand this reviewer's comments and agree that this information could improve the quality of the scientific information provided. Considering the main objective of the study – the screening and analysis of stress dimensions - both authors used a standardized RSS screening tool. This study is the first to map the situation in Slovakia with a focus on the next-of-kin carers of dementia sufferers. It is important to note that the authors suggested including focus group interviews in future research. Both authors have the ambition to continue collecting data on this topic. Cooperation between the two universities (Slovak and Norwegian) is starting with the Slovak Alzheimer's Society and there is a demand for research data. The first step was to collect data on the stress dimension; the second will be to collect data with a focus on resources, resilience, recovery or needs for carers through interviews.

There is no information with regards to psychiatric morbidity and medication of the participants. The authors state that caregiving is associated with depression and anxiety, so it would be expected a number of participants to have been treated for those disorders. It seems plausible that a person who had previously clinically relevant symptoms, possibly related to the caregiving status, may have score better in this scale, only because he/she receives and antidepressant or anxiolytic. Those issues should be commented by the authors.

We agree and have added two sentences in the limitation paragraph (lines 371-374).

In the results section (lines 169-171) the authors refer to unemployed carers. However, given the mean age of carers (>74 years) it would be expected that most would have been retired. Please clarify.

We deleted the sentence about unemployed informants. This is because we did not collect data on the number of hours spent on care per week. However, not only retired people participated in our study; there were also adults between the ages of 43 and 62 who are employed.

The most part of the first paragraph in the Discussion section (lines 253-259) should be removed to Methods section.

The first paragraph from the Discussion section was removed to the Methods section (now lines 119-125).

The authors state that this is the first study in their country with the use of the RSS. Are there other studies in Slovakia on the same topic that used other instruments? If so, you can make some general comparisons with your results.

In the Introduction (lines 64-72), both authors described the Slovak research (references 15-20) with a general link to this manuscript. Comparison is not possible due to the different focus of research during dementia care (end-of-life dementia care; burden of physical care or lack of knowledge of informal caregivers).

Best Regards, dr. Miroslava Tokovska on behalf of co-author Jana Solcova

Reviewer 2 Report

Thank you for the opportunity to review the manuscript “The Burden of Next-of-kin Carers of Dementia Sufferers in the Home Environment" (healthcare-2028637).

The authors designed a quantitative and descriptive study to screen different dimensions of burden experienced by next-of-kin carers for dementia sufferers in Slovakia.

The objectives and the rationale of the study are clearly stated.

The interpretation of results and study conclusions are supported by the data.

The Authors clearly emphasized the strengths of their study and clearly stated the limitations (page 8/9).

The study would benefit even more if the central results were summarized in a graph.

Author Response

Thank you for the opportunity to review the manuscript “The Burden of Next-of-kin Carers of Dementia Sufferers in the Home Environment" (healthcare-2028637).

The authors designed a quantitative and descriptive study to screen different dimensions of burden experienced by next-of-kin carers for dementia sufferers in Slovakia.

The objectives and the rationale of the study are clearly stated.

The interpretation of results and study conclusions are supported by the data.

The Authors clearly emphasized the strengths of their study and clearly stated the limitations (page 8/9).

The study would benefit even more if the central results were summarized in a graph.

We appreciate your positive feedback on our manuscript. We agree that it would be beneficial for readers to have central results in the graph. Based on suggestions from four other reviewers, we decided not to include this graph in our manuscript. We made all the necessary major revisions according to the proposals and provided each reviewer with reasons for accepting or rejecting the proposals. 

 Best Regards, dr. Miroslava Tokovska on behalf of co-author Jana Solcova

Reviewer 3 Report

The research topic is important and relatively new, but significant work needs to be undertaken to clarify the justification of the study.

The manuscript needs to be clarified what is significant, add some review sub-section, methods (measures), findings, discussion, and conclusion. The manuscript needs a major revision.

Abstract

The introduction of the abstract is not clear what is the main purpose (see line 11-13), methods (line 14-19), findings (line 19-26) and then provided more detail the conclusion and some implications.

Introduction

The introduction required to extensive work in the field. The significance of the paper is unclear. The objectives is what and how investigates. What does study exist? How the author test the study?

Page 1: line 30-44.

Page 2: line 54-63.

*** The objectives are not clearly identified what and how your study test? (See line 65-72).

*** The first question (RQ1 and RQ2) should be paraphrased. It is not reflected in quantitative methods (factors, variables, and items)  testing.

Review

It is not clear if would have no review section. The author should be added the definition, conceptualization, the effect of burden of next-of-kin carers and home environment. Should be clear how many factors, variables, and items are included in the testing.

Methods

The paper used inappropriate methods especially design (why is included scale), sample (this sub-section is not clear how did you recruit the sample), data collection (is it not clear what and how did you distribute questionnaire, sample questions, items and contracted the participants), and analysis is very not clear the methods of analyzing.

*** Why the questionnaire yielded quantitative data. My question is, most of your testing is quantitative data, how did you analyze?

See line 79-157.

Results

1. Descriptive is not clear (see line 162-175)

2. Table 2 is more detailed what values means.

3. Table 4 is limited in explaining what communicate for (see line 237-248)

*** The big problems of this section are never going back to answer the question. Should be clear, why the results are not answered two RQ?

Discussion

Discussion lack in-depth debate with theories with other scholars. I found many points are missing in the discussion section. Many points are overstatement or debate. Why you did not follow as the research question and then discussed with the main findings.

*** First, should be added introduction of discussion.

*** Second, the following discussion with the RQ and main findings.

*** Third, added the social and practical implications.

Conclusions

Conclusion unclear what main findings are. Should be clear concluded with the main findings. For instance, the authors stated "Several important findings are generated in this study". It is clear conclusion? (See line 369-390).

References

The text and the reference lists are incorrect. Should be strictly followed the MDPI format is required.

The length of paragraph

The paragraph is strictly long-length between 7-8 lines per each paragraph is required.

English and communication

Many words, terms, and grammar, and sentences are wrong meaning. Should provided professional editing is required.

Author Response

The research topic is important and relatively new, but significant work needs to be undertaken to clarify the justification of the study.

Both authors are grateful for the feedback.

The manuscript needs to be clarified what is significant, add some review sub-section, methods (measures), findings, discussion, and conclusion. The manuscript needs a major revision.

Abstract

The introduction of the abstract is not clear what is the main purpose (see line 11-13), methods (line 14-19), findings (line 19-26) and then provided more detail the conclusion and some implications.

We have rewritten the abstract and clarified the information for the readers.

Introduction

The introduction required to extensive work in the field. The significance of the paper is unclear. The objectives is what and how investigates. What does study exist? How the author test the study?

Page 1: line 30-44.

Page 2: line 54-63.

*** The objectives are not clearly identified what and how your study test? (See line 65-72).

*** The first question (RQ1 and RQ2) should be paraphrased. It is not reflected in quantitative methods (factors, variables, and items)  testing.

The introduction was rewritten according to the reviewer and also other reviewers (1,2,4,5). Also RQ1 and RQ2 were paraphrased to one research question (lines 80-81).

Review

It is not clear if would have no review section. The author should be added the definition, conceptualization, the effect of burden of next-of-kin carers and home environment. Should be clear how many factors, variables, and items are included in the testing.

We added a definition of the concept of family caregivers and its characteristics, as well as that of dementia and its characteristics and consequences when it comes to the need for support and care in the introduction (lines 29-38).

Methods

The paper used inappropriate methods especially design (why is included scale), sample (this subsection is not clear how did you recruit the sample), data collection (is it not clear what and how did you distribute questionnaire, sample questions, items and contracted the participants), and analysis is very not clear the methods of analyzing.

In part 2 ‘Materials and methods’ (2.1 Study design), the research design is described with a focus on clarifying the RSS screening tool. The next sections (2.2 - 2.5) contain all the necessary descriptions and explanations in connection with the research design and data processing. According to the other reviewers (1,2,4 and 5), this part of our manuscript was approved without comments.

*** Why the questionnaire yielded quantitative data. My question is, most of your testing is quantitative data, how did you analyze?

See line 79-157.

Researchers decided to use the RSS (Relatives Stress Scala) which was used in other countries - please read part - 4. Discussion in the first paragraph (lines 268-277). Data analysis is part 2.5 (lines 167-173).

Results

  1. Descriptive is not clear (see line 162-175)
  2. Table 2 is more detailed what values means.
  3. Table 4 is limited in explaining what communicate for (see line 237-248)

*** The big problems of this section are never going back to answer the question. Should be clear, why the results are not answered two RQ?

Due to the fact that we reformulated the RQ, we believe that the results are comprehensively interpreted for the readers.

Discussion

Discussion lack in-depth debate with theories with other scholars. I found many points are missing in the discussion section. Many points are overstatement or debate. Why you did not follow as the research question and then discussed with the main findings.

*** First, should be added introduction of discussion.

*** Second, the following discussion with the RQ and main findings.

*** Third, added the social and practical implications.

Based on input from four other reviewers, we made any necessary major revisions to the suggestions and provided each reviewer with reasons for accepting or rejecting the suggestions.

Conclusions

Conclusion unclear what main findings are. Should be clear concluded with the main findings. For instance, the authors stated "Several important findings are generated in this study". It is clear conclusion? (See line 369-390).

On the one hand, we agree with the reviewer that the first sentence in the conclusion was unclear. We deleted it. On the other hand, we tried to do all revisions according to five reviewers. The other reviewers had no suggestions to improve or clarify the conclusion. For this reason, we leave the conclusion without substantial modifications.

References

The text and the reference lists are incorrect. Should be strictly followed the MDPI format is required.

Thank you for your feedback, we checked and made corrections.

The length of paragraph

The paragraph is strictly long-length between 7-8 lines per each paragraph is required.

We understood and tried to make more paragraphs especially in the introduction.

English and communication

Many words, terms, and grammar, and sentences are wrong meaning. Should provided professional editing is required.

Thanks for the suggestion. Our manuscript has been double-checked by a native speaker from the UK. Considering that the other four reviewers had no suggestions for language proofreading in the English language and after agreement with the editor, it is our decision to have the text edited by another proofreader.

Best Regards, dr. Miroslava Tokovska on behalf of co-author Jana Solcova

Reviewer 4 Report

First of all, I congratulate the authors for the article presented.  It is an important and relevant topic today.

I believe that the introduction should be more grounded, defining well the concept of family caregivers and its characteristics, as well as that of dementia and its characteristics and consequences when it comes to the need for support and care.  These are issues that, although common sense, should be clarified in a scientific article.

I believe that the research would have been much richer if a mixed methodology had been used, through the use of focus groups, which undoubtedly would have given more information than the questionnaire, which is also appropriate.  Significant stress scores can be expected to be obtained in these cases, but this yields little information.  Interviews, case studies and life histories would undoubtedly enrich the results more than simple correlations.

Best regards

Author Response

First of all, I congratulate the authors for the article presented.  It is an important and relevant topic today.

Thanks a lot for the nice feedback.

I believe that the introduction should be more grounded, defining well the concept of family caregivers and its characteristics, as well as that of dementia and its characteristics and consequences when it comes to the need for support and care.  These are issues that, although common sense, should be clarified in a scientific article.

We added definitions according to these suggestions in the introduction.

I believe that the research would have been much richer if a mixed methodology had been used, through the use of focus groups, which undoubtedly would have given more information than the questionnaire, which is also appropriate.  Significant stress scores can be expected to be obtained in these cases, but this yields little information.  Interviews, case studies and life histories would undoubtedly enrich the results more than simple correlations.

We agree with the reviewer. In the last part of discussion (last sentence) authors wrote that future research should be conducted in dyadic interviews or focus group interviews. Unfortunately, it is not possible in this manuscript to collect new data through interviews. This study is the first to map the situation in Slovakia with a focus on the next-of-kin carers of dementia sufferers. Both authors have the ambition to continue collecting data on this topic. Cooperation between the two universities (Slovak and Norwegian) and with the Slovak Alzheimer's Society have started this year and there is a demand for research data collection. The first step was to collect data on the dimension of stress; the second will be to collect data with a focus on resources, resilience, recovery or the needs of carers through interviews. We are looking forward to continuing with publishing original articles in connection with this theme.

Best Regards, dr. Miroslava Tokovska on behalf of co-author Jana Solcova

Reviewer 5 Report

Dear Authors,

Congratulations on the developed article, the theme - burden experienced by next-of-kin carers for dementia sufferers in Slovakia - is relevant and current.

The method, although simple, presents conditions for discussion.

The sample is small and not probabilistic, which prevents the generalization of the results. Furthermore, it raises doubts about the relevance of the results. Therefore, I recommend adding some interviews to bring more remarkable contributions about what can be done to improve the respondents' quality of life. For example, what would these people like to see changed to make them feel less burdened?

The discussion is well developed, I make two points: 1. move the limitations to the conclusion, before the suggestions for future research. 2. discuss if there is a culture-related difference, in the results of similar research in other European countries - that were mentioned in the introduction.

I hope the comments are helpful and I wish the authors success.

Author Response

Dear Authors,

Congratulations on the developed article, the theme - burden experienced by next-of-kin carers for dementia sufferers in Slovakia - is relevant and current.

We appreciate your positive feedback on our relevant and current theme.

The method, although simple, presents conditions for discussion.

The sample is small and not probabilistic, which prevents the generalization of the results. Furthermore, it raises doubts about the relevance of the results. Therefore, I recommend adding some interviews to bring more remarkable contributions about what can be done to improve the respondents' quality of life. For example, what would these people like to see changed to make them feel less burdened?

We agree and understand the reviewer's suggestion. The authors did not have the intention of generalising the results. Adding some interviews could make this study more readable and interesting. Unfortunately, it is not possible to add it now. The authors have the following explanation for this: Considering the main goal of the study – screening and analysis of stress dimensions, both authors used a standardized RSS screening tool. This study is the first to map the situation in Slovakia with a focus on the closest caregivers of people suffering from dementia. It is important to note that the authors suggested including focus group interviews in future research. Both authors have the ambition to continue collecting data on this topic. Cooperation between two universities (Slovak and Norwegian) and the Slovak Alzheimer Society has started, and there is a demand for research data. The first step was to collect data on the stress dimension; the second will be data collection focused on resources, resilience, recovery or the needs of caregivers through interviews.

The discussion is well developed, I make two points: 1. move the limitations to the conclusion, before the suggestions for future research. 2. discuss if there is a culture-related difference, in the results of similar research in other European countries - that were mentioned in the introduction.

Regarding your moving of the limitations, we would like to leave it as the last paragraph in the discussion. We also reviewed other published articles in the MDPI Healthcare journal, and we noted that it is usually the last part of the discussion, not the conclusion. Thank you for understanding.

Culture-related differences are under-reported in our manuscript, we agree. Most researchers emphasize the “same” conclusion regarding dementia care: the mental and physical health burden of informal caregivers worldwide, women are more likely to be responsible for caring for their families and children, and are often also externally employed, leading to a double burden and increasing the likelihood of negative health effects that may result in psychiatric treatment. Moral responsibility is also typical for a collectivist country such as Slovakia. We think that these important conclusions from other countries are mentioned in the discussion. 

I hope the comments are helpful and I wish the authors success.

Thank you for the good suggestions!

Best Regards, dr. Miroslava Tokovska on behalf of co-author Jana Solcova

Round 2

Reviewer 1 Report

The manuscript could be accepted for publication in present form.

Reviewer 3 Report

Yes. All comments are revised and read well.

Reviewer 4 Report

I congratulate the authors for their efforts in responding to the proposed revisions.  Although I still find the study limited, I believe that it adequately justifies this fact, which is why I consider it sufficient for publication.

Best regards.